# Transcriptomic Analysis Reveals Key Roles of (p)ppGpp and DksA in Regulating Metabolism and Chemotaxis in *Yersinia enterocolitica*

**DOI:** 10.3390/ijms24087612

**Published:** 2023-04-20

**Authors:** Can Huang, Wenqian Li, Jingyu Chen

**Affiliations:** Beijing Laboratory for Food Quality and Safety, College of Food Science & Nutritional Engineering, China Agricultural University, 17 Qinghua East Rd., Beijing 100083, China

**Keywords:** *Yersinia enterocolitica*, (p)ppGpp, DksA, stringent response, chemotaxis

## Abstract

The stringent response is a rapid response system that is ubiquitous in bacteria, allowing them to sense changes in the external environment and undergo extensive physiological transformations. However, the regulators (p)ppGpp and DksA have extensive and complex regulatory patterns. Our previous studies demonstrated that (p)ppGpp and DksA in *Yersinia enterocolitica* positively co-regulated motility, antibiotic resistance, and environmental tolerance but had opposite roles in biofilm formation. To reveal the cellular functions regulated by (p)ppGpp and DksA comprehensively, the gene expression profiles of wild-type, Δ*relA*, Δ*relA*Δ*spoT*, and Δ*dksA*Δ*relA*Δ*spoT* strains were compared using RNA-Seq. Results showed that (p)ppGpp and DksA repressed the expression of ribosomal synthesis genes and enhanced the expression of genes involved in intracellular energy and material metabolism, amino acid transport and synthesis, flagella formation, and the phosphate transfer system. Additionally, (p)ppGpp and DksA inhibited amino acid utilization (such as arginine and cystine) and chemotaxis in *Y. enterocolitica*. Overall, the results of this study unraveled the link between (p)ppGpp and DksA in the metabolic networks, amino acid utilization, and chemotaxis in *Y. enterocolitica* and enhanced the understanding of stringent responses in *Enterobacteriaceae*.

## 1. Introduction

Bacteria utilize a series of sensory systems to monitor the external environment and regulate cellular metabolic networks in a timely manner to adapt to stressful conditions. Most of these systems convert external stimuli into regulation of the intracellular concentration of secondary messengers, which modulate key molecular targets and polymorphic phenotypes [1]. Guanosine-5′, 3′-pentaphosphate (pppGpp) and guanosine-5′, 3′-tetraphosphate (ppGpp) (collectively referred to as (p)ppGpp) are common secondary messengers that were originally found to induce a decline in rRNA and tRNA synthesis in response to amino acid starvation [2,3]. It is now known that several stress conditions, including both nutritional and environmental stimuli, induce an increase in intracellular (p)ppGpp concentration and lead to global changes in transcription profiles. This concentration-dependent reprogramming of cellular processes induced by (p)ppGpp is known as the stringent response.

The RelA-SpoT Homologue (RSH) controls the cellular pool of (p)ppGpp. In most bacteria, a single RSH enzyme with both synthetic and hydrolytic activities can use GTP or GDP and ATP to generate (p)ppGpp and can also hydrolyze (p)ppGpp to yield pyrophosphate and GDP or GTP. Monofunctional RelA and SpoT proteins are generally found in Betaproteobacteria and Gammaproteobacteria. RelA harbors the synthetic activity, whereas SpoT is mainly responsible for (p)ppGpp degradation and has weak synthase activity [3,4,5]. In addition, short single-domain RSH proteins, including small alarmone synthetases (SASs) and small alarmone hydrolases (SAHs), are present in various bacterial species [6,7].

(p)ppGpp is induced by various stresses and regulates intracellular metabolic levels and polymorphic bacterial phenotypes. Transcriptome analysis showed that approximately 16–26% of the genes regulated by the stringent response are involved in DNA, tRNA, fatty acid, amino acid, and membrane component synthesis [8,9]. Some studies have also reported impaired virulence, antibiotic resistance, and persistence in ΔRSH mutants [10]. There are three mechanisms of response for regulating (p)ppGpp. Th first is by altering the binding preference of RNA polymerase (RNAP) to the promoters of downstream target genes. (p)ppGpp molecules bind to two distinct sites on RNAP, namely the β’-ω interface (site 1) and DksA-β′ interface (site 2) [11,12,13]. They regulate the formation and decay rates of RNAP-promoter complexes as well as the escape of RNAPs from promoters, which may be dependent on the DNA sequence of the promoters, enabling sophisticated control over numerous genes. In addition, (p)ppGpp work by directly interacting with target proteins, such as DNA initiation enzyme (DnaG) [14], translation initiation factor 2 (IF2) [15], and elongation factor G (EF-G) [16]. Many of these identified proteins are GTP-binding enzymes, suggesting that (p)ppGpp may regulate their activity by competing with them for intracellular GTP [17]. (p)ppGpp can also indirectly regulate the expression of downstream target genes through transcription factors, such as σ^S^, cyclic di-AMP, and CodY [18,19,20]. These direct and indirect mechanisms and interactions with RNAP described above complete the global regulation of the stringent response on bacterial transcriptional and polymorphic phenotypes.

In addition to (p)ppGpp, the transcription factor DksA also plays a crucial role in the stringent response. According to bioinformatics analysis, DksA appears to be present in most Proteobacteria, but not in *Firmicutes* or *Thermophiles*. It should be emphasized that such a statement is not definitive owing to our limited knowledge of the protein structure of DksA. In fact, globular proteins similar to DksA that do not contain a zinc finger were found in *Pseudomonas aeruginosa* and *Rhodobacter sphaeroides* [21,22]. The DksA protein is composed of a globular domain and a long coiled coil, which can bind together with (p)ppGpp at the edge of the β′ subunit secondary channel of RNAP [23,24]. Generally, DksA is considered a molecular chaperone to enhance the signal of (p)ppGpp during a stringent response [2]. However, several studies have reported the independent and even antagonistic actions between DksA and (p)ppGpp. In *E. coli*, the transcriptional expression changes of flagella-related genes, such as *fliE*, *fliF*, and *fliG*, in DksA-deficient strains and (p)ppGpp-deficient strains show opposite trends and exhibit opposite phenotypes in motility and chemotaxis [9]. In addition, Lyzen et al. have demonstrated that the strength of the pArgX promoter is independently regulated by the DksA protein, which drives RNAP to form nonproductive complexes at the pArgX promoter [25]. Some studies have also shown that the phenotypic changes caused by (p)ppGpp deficiency can be compensated by overexpressing DksA [26]. These findings suggest a complex interplay between DksA and (p)ppGpp and deciphering their mode of action could provide insights into the regulatory mechanisms of stringent response and bacterial environmental adaptation.

We recently characterized a part of the physiological activity regulated by the stringent response factors (p)ppGpp and DksA in *Y. enterocolitica* [27]. It was found that (p)ppGpp and DksA synergistically regulate the motility, antibiotic resistance, and tolerance to oxidative stress of *Y. enterocolitica* but have independent or opposite effects on bacterial biofilm formation and tolerance to acid and a hyperosmotic environment. To fully elucidate the regulatory mechanisms of (p)ppGpp and DksA on gene expression, comparative transcription analyses were conducted using *Y. enterocolitica* wide-type strains and DksA-, (p)ppGpp-, and DksA(p)ppGpp-deficient strains. The roles of (p)ppGpp and DksA in the transport and utilization of arginine and cystine were verified using several experiments, and their positive effects on bacterial chemotaxis were revealed.

## 2. Results

### 2.1. (p)ppGpp- and DksA-Dependent Genes in Y. enterocolitica Identified Using RNA-Seq

Previous reports have shown that (p)ppGpp and DksA have extensive regulatory roles in *Escherichia coli* and *Xanthomonas citri* [9,10]. Our previous studies on *Y. enterocolitica* also showed that the regulatory patterns of (p)ppGpp and DksA at the phenotypic and genotypic levels are complex, showing both synergistic and antagonistic actions [27]. To better understand the function of (p)ppGpp and DksA in cellular processes in *Y. enterocolitica*, RNA-Seq was conducted to determine the transcription profiles of wild-type and mutant strains, including *Y. enterocolitica* Δ*dksA*, *Y. enterocolitica* Δ*relA*Δ*spoT*, and *Y. enterocolitica* Δ*dksA*Δ*relA*Δ*spoT*.

As shown in Appendix A, approximately 23.96 million, 26.06 million, 25.49 million, and 25.61 million raw reads were generated in WT, Δ*dksA*, Δ*relA*Δ*spoT*, and Δ*dksA*Δ*relA*Δ*spoT*, respectively, using Illumina sequencing. After removing low-quality reads, approximately 23.58 million, 25.69 million, 25.07 million, and 25.32 million clean reads were retained, respectively. The matching rate of these clean reads to the reference genome ranged from 97.72 to 98.15%, with a unique matching rate ranging from 93.64 to 95.92%. Saturation analysis showed that most clean reads were close to saturation at 40% of the mapping reads, indicating that the number of sequencing reads covered most of the expressed genes (Appendix A). In the coverage analysis, each curve had no biased peaks, indicating no sequencing bias (Appendix A). These results showed that the transcriptome sequencing results were reliable and could be used for subsequent bioinformatics analysis.

### 2.2. (p)ppGpp- and DksA- Dependent Cell Functions in Y. enterocolitica

The gene expression levels of the WT and mutant strains were analyzed using the RSEM software (Version 1.3.1). A total of 3651 genes were detected in WT, accounting for 84.65% of the total gene number in *Y. enterocolitica* (4313 genes). In addition, 3644 (84.49%), 3434 (79.62%), and 3614 (83.79%) genes were detected in Δ*dksA*, Δ*relA*Δ*spoT*, and Δ*dksA*Δ*relA*Δ*spoT*, respectively (Table 1). The DEGs with a |log_2_ fold change (FC)| > 1 and a *p*-adjust < 0.05 from mutant strains were analyzed. Among the 801 DEGs in Δ*dksA*, 387 genes were upregulated and 417 genes were downregulated. A total of 1437 DEGs were observed in Δ*relA*Δ*spoT*, including 695 upregulated genes and 742 downregulated genes. Deletion of both *dksA* and genes involved in (p)ppGpp production resulted in 560 upregulated and 499 downregulated genes. Principal component analysis (PCA) was performed to compare the samples. Three biological replicates of each strain clustered together to form distinct groups from the others, suggesting that major variation arose from differences between each strain (Figure 1A). Correlation analysis also showed that the differences in genes were mainly due to differences among the strains (Figure 1B).

To investigate the correlation between DksA and (p)ppGpp, a Venn analysis was performed to determine the co-expression and specific expression of DEGs. As shown in Figure 1C, more than 62% of DEGs in Δ*dksA* were regulated by (p)ppGpp, lending support to the claim that DksA acts as a pivotal cofactor of (p)ppGpp in global regulation. Furthermore, functional categories of the DEGs were categorized according to the KEGG database, and the pathways modulated by DksA and (p)ppGpp were similar, including carbon metabolism, amino acid metabolism, and membrane transport (Appendix A). Notably, 17, 14, and 15 cellular processes were significantly (*p*-adjust < 0.05) enriched in Δ*dksA*, Δ*relA*Δ*spoT*, and Δ*dksA*Δ*relA*Δ*spoT*, respectively, via the KEGG database (Figure 2). While DksA and (p)ppGpp regulate several similar cellular processes, such as ribosomes, amino acid metabolism, and cell metabolism, some distinct cellular processes are uniquely regulated by (p)ppGpp or DksA, revealing separate regulatory mechanisms. Taken collectively, these bioinformatic findings suggest that DksA and (p)ppGpp modulate comparable genes and cellular processes in *Y. enterocolitica*.

To further verify the accuracy of the RNA-Seq results, 19 DEGs from the mutant strains were randomly selected, and their expression levels were measured and compared to those of the WT using RT-qPCR. These genes are involved in biological processes enriched in bioinformatic analysis, such as translation (*rpsN*), flagellar formation (*fliE* and *flgC*), nutrient import (*gntT* and *hisJ*), RNA biosynthesis (*trmD* and *dusB*), and others. The amplified band’s uniqueness was confirmed through agarose electrophoresis (Appendix A). As expected, the expression levels of these candidate DEGs in Δ*dksA*, Δ*relA*Δ*spoT*, and Δ*dksA*Δ*relA*Δ*spoT* were consistent with the RNA-Seq sequencing results (Figure 3), with correlation coefficients of 0.96, 0.97, and 0.94, respectively. This indicates that the RNA-Seq results were reliable and accurate (Appendix A).

### 2.3. DksA and (p)ppGpp Inhibit Ribosomal Synthesis and Regulate Metabolic Networks

The main hallmark of the stringent response is the suppression of ribosome production. In Δ*dksA* and Δ*relA*Δ*spoT*, the expression of 43 and 50 ribosomal DEGs, respectively, was significantly increased (FC > 2 and *p*-adjust < 0.05). Of these DEGs, 42 were found to be jointly controlled by both DksA and (p)ppGpp (Appendix A), implying a shared regulatory mechanism. Notably, the deletion of *dksA*, *relA*, and *spoT* led to lower expression levels of 30 DEGs involved in ribosomal synthesis compared to Δ*dksA* or Δ*relA*Δ*spoT* alone. These results indicate that the regulatory pattern of ribosome synthesis is realized through site 2 of RNAP, which requires both (p)ppGpp and DksA.

Previous research had highlighted the essential role of (p)ppGpp or DksA in the growth of *Erwinia amylovora* and *Salmonella enterica* [28,29]. In this study, pathway enrichment analysis revealed that the TCA cycle, fatty acid degradation, and carbon fixation pathways of Δ*dksA*, as well as the purine metabolism pathway of Δ*relA*Δ*spoT* were significantly inhibited (Figure 4 and Figure 5). These pathways are involved in carbon source utilization and likely contribute to the slower growth of mutant strains in LBNS medium (Figure 6A). Our analyses thus suggest that both DksA and (p)ppGpp act to regulate metabolic networks while inhibiting genes involved in ribosome biosynthesis.

### 2.4. DksA and (p)ppGpp Are Required for Amino Acid Transport and Utilization

Previous studies have demonstrated that a stringent response could be triggered by amino acid starvation, and high cellular concentrations of (p)ppGpp and DksA have been shown to enhance amino acid biosynthesis and promote the expression of relevant transporters [30,31]. In this study, arginine and proline metabolism, as well as transporters for arginine, histidine, and glutamate, were repressed in Δ*dksA* (Figure 4). In addition, amino acid transporters for cystine, arginine, glutamate, and histidine were repressed in Δ*relA*Δ*spoT* (Figure 5). This suggests that restricted amino acid bioavailability could limit the growth of mutant strains. To confirm this, we assessed the growth of these strains in LBNS supplemented with virous amino acids (Figure 6). At the 30th hour (stationary phase) of incubation, the OD_600_ values of Δ*dksA*, Δ*relA*Δ*spoT*, and Δ*dksA*Δ*relA*Δ*spoT* were 17.6%, 33.50%, and 32.9%, respectively, which were lower than that of WT. However, we observed a reduction in the growth gaps after adding arginine, cystine, and histidine (Appendix A). The addition of cystine reduced the growth gap of Δ*dksA* from 17.61% to 11.79%, while arginine effectively reduced the growth gap in Δ*relA*Δ*spoT*. Of note, glutamic acid (3.94 g/L) had an inhibitory effect on the growth of *Y. enterocolitica*. Overall, these results suggest that (p)ppGpp and DksA are positive regulators for amino acid transport and utilization (such as arginine and cystine) in *Y. enterocolitica*.

### 2.5. DksA and ppGpp Positively Affect Bacterial Chemotaxis

The biosynthesis of flagella is a vital process that requires a significant amount of resources and energy, enabling bacteria to achieve motility, competitiveness, and virulence [32,33]. Results obtained from RNA-Seq analysis indicated that in the absence of DksA, the expression of most genes involved in flagellar synthesis was elevated, except for *fliC*, *fliC2*, and *fliC3* (Figure 4). In our previous study, we confirmed that the motility and flagella formation abilities in Δ*dksA* were repressed. Combined with this phenotypic result, DksA is determined to be the key factor in filament synthesis. Although flagellar assembly in Δ*relA*Δ*spoT* was not a significant functional group in the KEGG enrichment analysis (*p*-value = 0.2584), 15 genes involved in the flagellar assembly were perturbed by (p)ppGpp knockout. Among these genes, only *flhD* was significantly upregulated (FC > 2), whereas the remaining genes (*fliN*, *flgI*, *flgH*, *flgE*, *flgD*, *fliQ*, *fliS*, *fliT*, *flgG*, *flgF*, *fliJ*, *fliK*, *fliR*, and *fliC*) were significantly downregulated (FC < 2). The decreased expression of these flagellar genes may be responsible for the decreased motility and flagellar formation ability of Δ*relA*Δ*spoT* [27]. In addition, the expression levels of these genes in Δ*dksA*Δ*relA*Δ*spoT* were more similar to those in Δ*dksA* than in Δ*relA*Δ*spoT*, indicating that DksA plays a more critical role in the regulation of flagellar biosynthesis during stringent response.

Flagellar synthesis is critical for the chemotactic ability of bacteria to move to a more favorable environment and confer a selective advantage. The Rcs phosphorelay system was found to be overexpressed in the functional group of Δ*dksA*, which inhibits the chemotactic ability of *Y. enterocolitica* [34]. These studies imply that a stringent response has a significant regulatory effect on the chemotactic ability of *Y. enterocolitica*. To validate this, the chemotaxis ability of the mutant strains was evaluated using swimming and competitive capillary assays. As shown in Figure 7, the swimming diameter of the mutant strain was smaller than that of the WT for the three carbon sources of glucose, glycerol, and malic acid. Moreover, the mutant strains accounted for 26.7% or less of the total number of colonies after incubation (Figure 7C). The experimental results demonstrate that DksA and (p)ppGpp play positive roles in *Y. enterocolitica* motility, flagella formation ability, and chemotaxis.

## 3. Discussion

The stringent response is a sensory system widely present in bacteria that can regulate various cellular processes such as fatty acid metabolism and flagellar synthesis, thereby determining bacterial survival, virulence, and pathogenicity [35,36]. This system works by altering the concentration of the intracellular secondary messenger (p)ppGpp with the participation of DksA. Although studies have confirmed that DksA is the main cofactor of (p)ppGpp during stringent response, (p)ppGpp- and DksA-deficient strains display differences at the genotypic and phenotypic levels [25,37]. Our previous studies also showed that DksA and (p)ppGpp cooperatively regulate antibiotic resistance and environmental tolerance in *Y. enterocolitica*, whereas they exhibit dependent or independent roles in biofilm synthesis [27]. To comprehensively understand the regulatory function of the stringent response in *Y. enterocolitica*, WT and the mutant strains Δ*dksA*, Δ*relA*Δ*spoT*, and Δ*dksA*Δ*relA*Δ*spoT* were subjected to comparative transcriptome analysis and phenotypic characterization. Our study provides new insights into the regulation of ribosomal synthesis, energy metabolism, transporters, flagellar synthesis, two-component systems, and flagellar synthesis by (p)ppGpp and DksA in *Y. enterocolitica*.

The interplay between (p)ppGpp and DksA has been widely explored in *E. coli* and *X. citri* [9,10]. DksA and (p)ppGpp bind together in the secondary channel of RNAP and positively or negatively affect the transcriptional activity of RNAP according to the properties that are intrinsic to the promoter [24]. In addition, (p)ppGpp has also been confirmed to directly bind to the promoters of the genes involved in amino acid biosynthesis, and DksA can also form a complex with DNAJ in response to external redox signals, indicating that (p)ppGpp and DksA also have regulatory effects independent of each other [38,39]. In the present study, 18.57% of the total genes of *Y. enterocolitica* were regulated by DksA, while 33.32% of the total genes of Δ*relA*Δ*spoT* were regulated, showing comprehensive and extensive regulation of the stringent response (Table 1). Among these genes, 496 were covered by DksA and (p)ppGpp, accounting for 61.92% of the total DEGs in Δ*dksA*, indicating that the main function of DksA is to participate in (p)ppGpp. In addition, DksA and (p)ppGpp exhibited negative interplay with certain DEGs (such as the flagella synthesis gene *fliE*). Similar results were also reported by Zhang et al., who found that the deficiency of DksA and (p)ppGpp resulted in the different expression of the siderophore synthesis and utilization gene cluster in *X. citri* [10]. Identifying the interaction between DksA and (p)ppGpp and dissecting its influence on bacterial physiological activities requires further investigation.

During the stringent response, cellular energy and material are rapidly reallocated from the synthesis of macromolecules for reproduction to produce factors that are crucial for survival. Reports have shown that (p)ppGpp or DksA is required for the growth of *E. coli* and *S. enterica* [36,40]. Our previous studies also showed that the maximum growth values of Δ*dksA* and Δ*relA*Δ*spoT* in LBNS medium were weaker than those of the WT strain, and Δ*relA*Δ*spoT* cannot even grow in M63 minimal medium (a minimum medium, an oligotrophic environment for bacteria) [27]. Due to the complexity of growth-related factors, it is difficult to explain the growth defect with a few specific cellular processes. Nevertheless, the decline in cellular carbon metabolism and amino acid utilization cannot be ignored. From the enrichment analysis of DEGs, the knockout of DksA or (p)ppGpp resulted in decreased expression of almost all genes in the tricarboxylic acid cycle (TCA cycle), thus reducing the level of intracellular material and energy metabolism, including the fatty acid metabolism and arginine synthesis pathways. However, the decreased expression level of amino acid transporters in the mutant strains also made it difficult to absorb external amino acids (Figure 4 and Figure 5). To verify that the limitation in amino acids is a possible factor for the growth defects, the growth condition of *Y. enterocolitica* in LBNS medium supplemented with single or mixed amino acids was measured. The results showed that the addition of arginine reduced the growth gap between the WT and the Δ*relA*Δ*spoT* strain in the stationary phase from 33.50% to 9.11% (Appendix A), indicating that (p)ppGpp is required for arginine transport and utilization in *Y. enterocolitica*. In addition, cystine utilization may respond to the defective growth in Δ*dksA*, which could minimize the growth gap with the WT.

Chemotaxis, the motility response to chemical stimuli in the external environment, is critical for bacteria to find food and stay away from harmful environments [41]. The flagella are the main motility organ of bacteria, and their synthesis is the basis of bacterial chemotaxis [42,43]. However, flagellar synthesis is an extremely energy-intensive process that requires approximately 2% of the energy in *E. coli* [44]. Knockout of (p)ppGpp and DksA leads to a decrease in energy and substance metabolism. Therefore, (p)ppGpp and DksA are likely to regulate bacterial flagellar synthesis and bacterial chemotaxis. It has been reported that flagellar biosynthesis was inhibited in both Δ*dksA* and Δ*relA*Δ*spoT*. Interestingly, in Δ*dksA*, only the expression of *fliC* decreased, whereas the expression of other synthetic genes increased or did not change significantly, indicating that DksA inhibits flagellar biosynthesis by repressing filament synthesis (Figure 4). Although most of the genes related to flagellar synthesis were downregulated in Δ*relA*Δ*spoT*, these genes in Δ*dksA*Δ*relA*Δ*spoT* exhibited a similar trend in Δ*dksA*. This may be due to the critical role of DksA in regulating the synthesis of flagella in *Y. enterocolitica* than (p)ppGpp and the fact that an independent regulatory mechanism exists between them. Because of the impaired flagellar synthesis, it is not surprising that the mutant strains have decreased chemotaxis. Notably, increased expression of the regulator of capsule synthesis (RCS) phosphorelay system was observed, which has been reported to reduce the chemotaxis of *Y. enterocolitica* in glucose, glycerol, and malate. Swimming and competition experiments in this study obtained similar results (Figure 6), indicating that an indirect mechanism may be involved in the regulation of bacterial flagellar synthesis genes by DksA.

In summary, this study revealed the regulation of stringent response factors (p)ppGpp and DksA on gene expression and cell function in *Y. enterocolitica*, focusing on the role of stringent response in bacterial ribosome synthesis and cell metabolism. Results showed that DksA and (p)ppGpp could positively promote the utilization and transport of amino acids, as well as bacterial flagellar synthesis and chemotactic ability. Since these phenotypes are critical for bacterial viability, colonization, and virulence, these findings demonstrated that DksA and (p)ppGpp could be potential targets for developing new disinfectants or agents.

## 4. Material and Methods

### 4.1. Bacterial Strains and Growth Conditions

The strains used in this study are listed in Table 2 and constructed based on previous studies [27]. *Y. enterocolitica* ATCC23715 (biotype 1B and serotype O:8) is a wild-type strain (WT), which is the basis of the mutant strains Δ*dksA*, Δ*relA*Δ*spoT*, and Δ*dksA*Δ*relA*Δ*spoT*, and the complementary strains Δ*dksA*(*dksA*) and Δ*relA*Δ*spoT*(*spoT*). All strains were grown at 26 °C in LB medium without salt (defined as LBNS) containing 5 g/L yeast extract and 10 g/L tryptone. For induction of the *dksA* gene and the *spoT* gene, complement strains were cultured in the medium supplemented with 0.02 g/L L-arabinose. In addition, 100 μg/mL of ampicillin and CIN (15 μg/mL of cefsulodin, 4 μg/mL of irgasan, and 2.5 μg/mL of novobiocin) were added to the growth medium when required.

### 4.2. RNA Extraction, Library Construction, and Sequencing

Fresh bacterial cultures of each strain were inoculated into 50 mL of LBNS medium at 1% and cultured to the mid-logarithmic phase (OD_600_ = 0.4). Each strain was carried out in three biological replicates. Bacterial cells were harvested and used to extract the total RNA using TRIzol^®^ Reagent according to the manufacturer’s instructions (Invitrogen). A high-quality RNA sample (OD_260/280_ = 1.8–2.0, OD_260/230_ ≥ 2.0, RIN ≥ 6.5, 28S:18S ≥ 1.0) was used to construct the sequencing library. 

A RNA-Seq transcriptome library was performed using a TruSeq^TM^ RNA sample preparation kit from Illumina (San Diego, CA, USA). After a cluster generation, a paired-end RNA-Seq sequencing library was sequenced with the Illumina HiSeq × TEN (2 × 150 bp read length). The processing of original images to sequences, base-calling, and quality value calculations were performed using the Illumina GA Pipeline (version 1.6), in which 150 bp paired-end reads were obtained.

### 4.3. Gene Annotation and Data Analysis

To ensure the accuracy of the data, the raw data is first filtered to obtain high-quality, clean data. Additionally, Bowtie2 (http://bowtie-bio.sourceforge.net/bowtie2/index.shtml accessed on 14 January 2021) was used to align the clean data of each sample with the reference genome (https://www.ncbi.nlm.nih.gov/genome/genomes/1041 accessed on 14 January 2021) and the gene expression level was measured by RSEM (http://deweylab.github.io/RSEM/ accessed on 14 January 2021), yielding transcripts per million reads (TPM). For each dataset, DESeq (http://www.bioconductor.org/packages/release/bioc/html/DESeq.html accessed on 14 January 2021)was used to analyze and identify the differences in genes among the samples. The significantly differentially expressed genes (DEGs) were identified with the inclusion criteria of |FC| > 2 and a *p*-adjust < 0.05. The transcriptome data were submitted to NCBI’s sequence read archive (SRA) database, and the SRA accession number is PRJNA943099. For the Kyoto Encyclopedia of Genes and Genomes (KEGG) enrichment analysis of DEGs, KOBAS 2.0 (http://kobas.cbi.pku.edu.cn accessed on 14 January 2021) was used to identify a statistically significantly enriched pathway using Fisher’s exact test. To control calculated false positive rates, *p*-values were corrected using 4 multiple testing methods (Bonferroni, Holm, Sidak, and the false discovery rate), and the pathways with a *p*-adjust < 0.05 were chosen.

### 4.4. RT-qPCR

Nineteen random DEGs were selected to verify the accuracy of the RNA-Seq data by RT-qPCR, and all the primers were designed using Primer Premier 6 software (Appendix A). The extracted total RNA was used to perform cDNA synthesis using the PrimeScriptTM RT Master Mix (Takara, Japan) according to the manufacturer’s instructions. RT-qPCR reactions were performed using TB Green^®^ *Premix Ex Taq*™ II (Takara), and all the reactions of each strain were performed for at least three biological replicates. The 16sRNA gene in *Y. enterocolitica* was used as an endogenous control, and the relative transcription of the target genes was calculated using the 2^-ΔΔCt^ method previously described [45].

### 4.5. In Vitro Bacterial Growth

The single clones of each strain were inoculated into 5 mL of LBNS liquid medium and grown overnight at 28 °C, 180 rpm. The cultures were then diluted into fresh LBNS liquid medium and supplemented with one single amino acid or multiple amino acids to an initial OD_600_ = 0.02. In specific experiments, the single amino acids equivalent to the nitrogen content of 1.5 g/L NH_4_Cl were added to the medium. The final concentration of cystine was selected as 0.1 g/L, as it has a low solubility in water. In the mixed amino acid solution, the nitrogen content of each amino acid is equal, and their total nitrogen content is equivalent to 1.5 g/L NH_4_Cl. A quantity of 200 μL of diluted culture was added to each well of the 100-well plate, and then the growth of *Y. enterocolitica* was monitored at 600 nm using an automatic curve growth instrument every 1 h. The experiment was performed in five biological replicates at least twice.

### 4.6. Swim Plate and Competitive Capillary Assays for Chemotaxis

For swim plate assays, strains were grown to mid-logarithmic phase in LBNS medium at 26 °C, and the bacteria cells were harvested and washed twice with chemotaxis buffer (10 mM KH_2_PO_4_, 10 mM KH_2_PO_4_, and 0.1 mM EDTA). Then, the cells were suspended at OD_600_ = 0.5 using the chemotaxis buffer. A quantity of 5 µL of bacterial suspension was inoculated into the center of the semi-solid LBNS plates (0.35% agar) containing 10 mM of different carbon sources (including glucose, malic acid, and glycerol). The plates were incubated at 26 °C for 48 h without being inverted, and the diameters and images of the chemotactic ring were measured and captured.

For competitive capillary assays, the mid-logarithmic phases of the bacterial cultures were harvested, washed twice with PBS, and then diluted to OD_600_ = 0.01. Aliquots of WT and mutant strains (Δ*dksA*, Δ*relA*Δ*spoT*, and Δ*dksA*Δ*relA*Δ*spoT*) were mixed, and 200 μL of the bacterial mixture was added to each well of a 96-well plate. One end of the capillary is heat sealed by a flame, and the other end is immediately inserted into the chemoattractant (10 mM of glucose, malic acid, or glycerol) or control (PBS). When the liquid height was stabilized at about 1 cm, the capillary was transferred to the 96-well plate containing the mixed bacterial solution and incubated at 26 °C for 1 h. Subsequently, the liquid in the capillary was removed and diluted in 1 mL of PBS. The collected bacteria were diluted with PBS and plated on the LBNS solid culture [29]. The plate was incubated at 26 °C for 36 h, and the number of colonies was counted, and PCR using colonies as a template was performed using specific primers (Appendix A) to determine the ratio of the two strains in the capillary.

### 4.7. Statistical Analysis

Statistical analysis was performed using one-way analysis of variance with Dunnett’s multiple comparison test * *p* < 0.05, ** *p* < 0.001, and *** *p* < 0.0001.

## Figures and Tables

**Figure 1 ijms-24-07612-f001:**
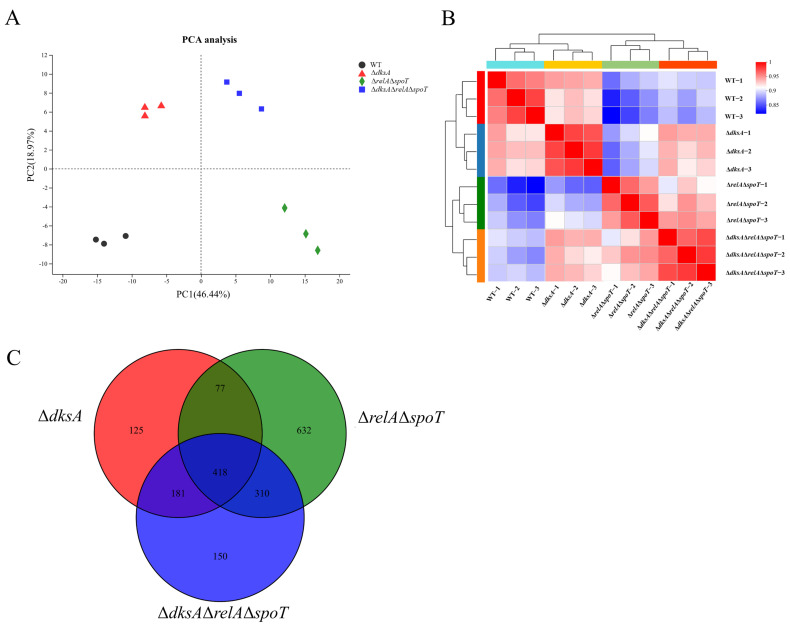
Expression analysis between samples. (**A**) Principal component analysis of twelve samples. (**B**) Correlation analysis of twelve samples. (**C**) Venn diagram of overlapping DEGs between Δ*dksA*, Δ*relA*Δ*spoT*, and Δ*dksA*Δ*relA*Δ*spoT*.

**Figure 2 ijms-24-07612-f002:**
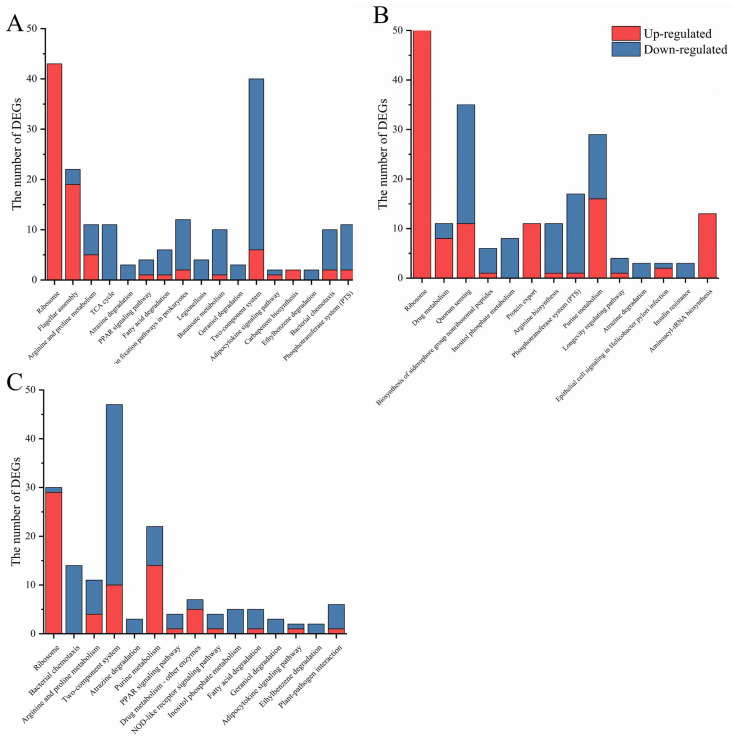
Functional enrichment analysis of DEGs based on the KEGG database. (**A**) Δ*dksA*; (**B**) Δ*relA*Δ*spoT*; and (**C**) Δ*dksA*Δ*relA*Δ*spoT*.

**Figure 3 ijms-24-07612-f003:**
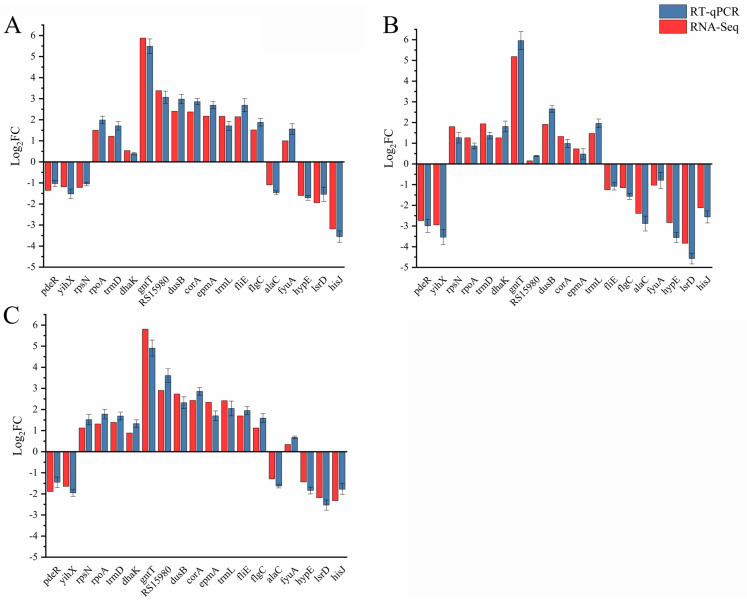
Validation of candidate DEGs by RT-qPCR. (**A**) Gene transcription in Δ*dksA*. (**B**) Gene transcription in Δ*relA*Δ*spoT*. (**C**) Gene transcription in Δ*dksA*Δ*relA*Δ*spoT*. The data are the means and SEM from three independent RT-qPCRs.

**Figure 4 ijms-24-07612-f004:**
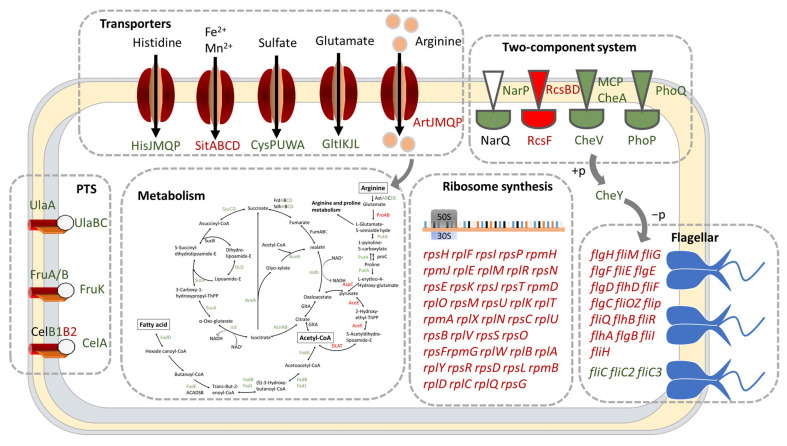
The model for the cellular processes of Δ*dksA*. Red, green, and black indicate significant upregulation (FC > 2 and *p*-adjust < 0.05), significant downregulation (FC < 0.5 and *p*-adjust < 0.05), and no significant regulation (0.5 ≤ FC ≤ 2 or *p*-adjust ≥ 0.05), respectively.

**Figure 5 ijms-24-07612-f005:**
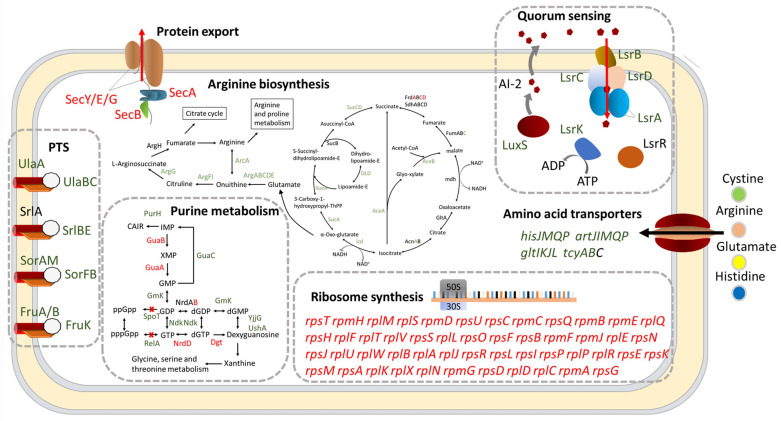
The model for the cellular processes of Δ*relA*Δ*spoT*. Red, green, and black indicate significant upregulation (FC > 2 and *p*-adjust < 0.05), significant downregulation (FC < 0.5 and *p*-adjust < 0.05), and no significant regulation (0.5 ≤ FC ≤ 2 or *p*-adjust ≥ 0.05), respectively.

**Figure 6 ijms-24-07612-f006:**
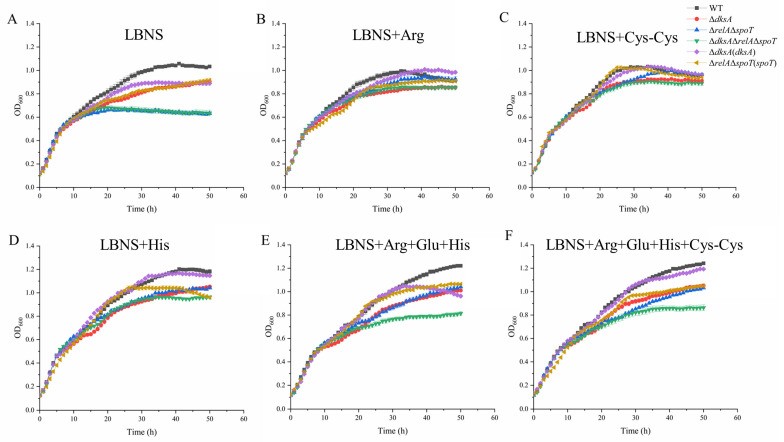
Growth characteristics of WT and mutant strains in liquid medium. (**A**) LBNS; (**B**) LBNS supplemented with 1.16 g/L arginine; (**C**) LBNS supplemented with 0.1 g/L cystine; (**D**) LBNS supplemented with 1.28 g/L histidine; (**E**) LBNS supplemented with 0.38 g/L arginine, 1.31 g/L glutamic acid, and 0.42 g/L histidine; (**F**) LBNS supplemented with 0.29 g/L arginine, 0.1 g/L cystine, 0.98 g/L glutamic acid, and 0.32 g/L histidine. The data are the means OD_600_ for five independent cultures and the standard errors of the means.

**Figure 7 ijms-24-07612-f007:**
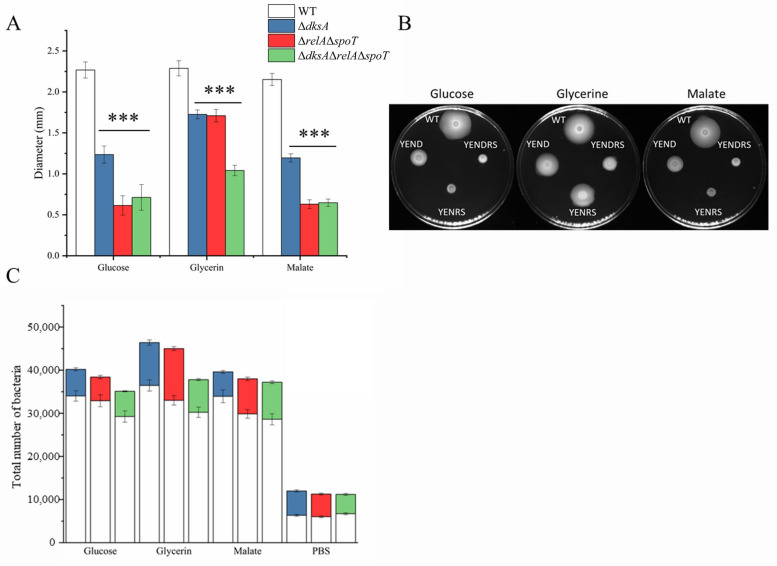
Comparison of the chemotactic responses of WT and mutant strains in three carbon resources. (**A**) Quantification of swim diameters. (**B**) Images of a swim plate. (**C**) Competitive chemotactic responses of WT and mutant strains. The data are presented as the mean ± SD of at least three biological repeats, and the error bars indicate standard deviations. An asterisk indicates a significant difference (*** *p* < 0.0001).

**Table 1 ijms-24-07612-t001:** Statistical analysis of DEGs.

Samples	Genetic Number	Ratios	Number of DEGs	DEGs	
				Upregulated	Downregulated
WT	3651	84.65%	0	0	0
Δ*dksA*	3644	84.49%	801	384	417
Δ*relA*Δ*spoT*	3434	79.62%	1437	695	742
Δ*dksA*Δ*relA*Δ*spoT*	3614	83.79%	1059	560	499

**Table 2 ijms-24-07612-t002:** Strains and plasmids used in this study.

Strain	Genotype	Sources
WT	*Y. enterocolitica* ATCC 23715; Serotype O:8; Biotype 1B; pYV-	Lab stored
Δ*dksA*	*Y. enterocolitica* ATCC 23715 derived, Δ *dksA*	Lab stored
Δ*relA*Δ*spoT*	*Y. enterocolitica* ATCC 23715 derived, Δ *relA* Δ *spoT*	Lab stored
Δ*dksA*Δ*relA*Δ*spoT*	Δ*relA*Δ*spoT* derived, Δ *dksA*	Lab stored
Δ*dksA*(*dksA*)	Δ*dksA* harboring pBAD24—*dksA*, AMPr	Lab stored
Δ*relA*Δ*spoT*(*spoT*)	Δ*relA*Δ*spoT* harboring pBAD24—*spoT*, AMP^r^	Lab stored

## Data Availability

The transcriptome data have been submitted to NCBI’s sequence read archive (SRA) database under accession number PRJNA943099.

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
