# Peer review of "Transcriptomic Analysis Reveals Key Roles of (p)ppGpp and DksA in Regulating Metabolism and Chemotaxis in Yersinia enterocolitica"

_ijms, 2023, doi:10.3390/ijms24087612_

Round 1

Reviewer 1 Report

The author's have prepared and submitted an interesting manuscript which further elucidates the roles of regulators of metabolic network in Y. enterocolitica.  The manuscript adds new information for the readers interested in this subject.  While the manuscript is sound it suffers from some issues which affect readability and comprehension.  These minor issues should be addressed by the authors:

1. Line 181, define the term DEG (assumed to mean Database of Essential Genes).

2. Lines 267-271, in discussion data in Table S2, please clarify the units associated with the numbers (BP?).

3.  Line 288, the authors refer the reader to Table 2.  However, no Table 2 data were included in the manuscript, only a Table header.  Please add these data.

4.  Line 383, change the period after YENRS to a comma.

5.  Line 418, "Figure 6" should be changed to "Figure 7 A/B", and LINE 422 "Figure 6C" should be changed to "Figure 7C".

6.  Line 438, remove the word "in" at the end of this line.

7.  Line 451, the phrase " thereby altering its the target genes" should be reworded to improve clarity.

8.  Line 484, add a "d" to the word "decrease".

9.  Line 494, remove the word "most" at the end of the line.

10.  Lines 539-540, The author's statement "these findings demonstrated that the stringent response inhibited the pathogenicity of Y. enterocolitica" is not substantiated by the data provided as no infection studies were included in the manuscript.  This phrase should be either eliminated or reworded.

Reviewer 2 Report

Comments on the manuscript “Transcriptomic analysis reveals the positive roles of (p)ppGpp and DksA in regulating the metabolic network, amino acid utilization, and chemotaxis in Yersinia enterocolitica

In the manuscript “Transcriptomic analysis reveals the positive roles of (p)ppGpp and DksA in regulating the metabolic network, amino acid utilization, and chemotaxis in Yersinia enterocolitica”, Huang and colleagues establish the transcriptomic landscape of the (p)ppGpp and DksA-mediated stringent response in Y. enterocolitica.  The paper does not provide much molecular insights on (p)ppGpp and DksA-mediated gene regulations, but rather offers resources for researchers investigating stress response in Y. enterocolitica.  

While the experiments are generally well conducted, the poor quality of the writing and of the data presentation make the reading of the article particularly unpleasant.  This manuscript should be considered for publication only if the authors seriously improve the presentation and the writing. 

Comments

1)    Title. The authors should change the title of the manuscript for something like “Transcriptomic analysis reveals key roles of (p)ppGpp and DksA in regulating metabolism and chemotaxis in Yersinia enterocolitica

2)    The English of the manuscript must be improved and would greatly benefit from professional editing. Many parts of the manuscript are quite hard to understand due to poor writing. 

3)    Highlights. First point. Lines 37-38. 

Gene expression profiles cannot be “sequenced”.  This phrasing is scientifically inaccurate.  Authors should instead write that gene expression profiles were “established” or “compared” between WT and mutant strains, and also mention under which conditions the transcriptomic study was performed. 

4)    Introduction. Lines 110-111. “[…] some studies have shown that gene regulation by DksA and (p)ppGpp is not completely consistent”.

What do the authors mean by “gene regulation by DksA and (p)ppGpp is not completely consistent”?  That the two proteins’ regulons are not completely overlapping?  The authors should develop more here. 

5)    Introduction. Lines 115-116. “[…] Meanwhile, (p)ppGpp and DksA play a direct influence mechanism”

This sentence is not well formulated.  Do the authors mean that (p)ppGpp and DksA have direct effect on the RNA polymerase? 

6)    Introduction. Lines 122-123. “However, the regulatory mechanisms at the genotypic and phenotypic are extremely complex”

Do the authors mean that it is sometimes hard to ascribe a given (p)ppGpp or DksA-mediated regulation a given phenotype?  The sentence has to be rephrased.

7)    Results. Section titles. Authors should aim at using take-home messages summarizing the findings in each section of the Results instead of descriptions of the experimental work that was performed (e.g. “Validation of RNA-Seq accuracy using RT-qPCR”).    

8)    Figures. The quality of the figures is in general rather poor, making it difficult for the reader to discern any information. The font size should be increased in most of the figures to make the text readable in the printed version of the manuscript. The image quality should also be increased as most of the figures are quite pixelated.

9)    The authors should refrain using the YEND, YENRS, YENDRS, YEND-D, YENRS-S notations in the text and figure as it requires the reader to constantly remember the genotypes associated with each abbreviation.  The genotype of each strain should instead be used to define them in the text or figures.

10) Discussion. Lines 534-542. “Results showed that DksA and (p)ppGpp could positively promote the utilization and transport of amino acids, as well as the bacterial flagella synthesis and the chemotactic ability. Since these phenotypes are critical for bacterial viability, colonization and virulence, these findings demonstrated that the stringent response inhibited the pathogenicity of Y. enterocolitica  

This part of the manuscript seems to contain two contradictory statements, the authors first claiming that DksA and (p)ppGpp contribute to phenotypes that are critical for bacterial virulence and then that the stringent response inhibited the pathogenicity of Y. enterocolitica.  The authors should further clarify their statements. 

Minor comments 

-       Abstract. Lines 21-22. “[…] compared the gene expression profiles of wild-type and mutant strainsusing RNA-Seq […]”. 

The authors should clearly mention in the sentence that the mutant strains they are referring to are (p)ppGpp synthesis and dksA mutants. 

-       Introduction. Lines 47-49. “Most of these systems convert external stimuli into intracellular concentrations of secondary messengers […]”.

The statement is inaccurate.  Authors should rephrase the sentence by stating that “Most systems convert external stimuli into regulation of the intracellular concentration of secondary messengers”. 

-       Introduction. Lines 74-77. “Transcriptome results show that approximatively 16-26% of genes areregulated by stringent responses involved in the synthesis of DNA […]

This sentence should be rephrased.  “The” should be inserted between “of” and “genes”, and “are” should be removed between “genes” and “regulated”.  The word “response” should be singular, and “are” inserted between “response” and “involved”. 

-       Introduction. Lines 107. “[…] but do not containing” 

Replace “do not” with “that are not”. 

-       Results. Lines 263-266. “The quality test showed that the total RNA had RNA integrity number […]” 

Such details should be moved to the Material and Methods section. 

-       Results. Lines 260-261. “DrelADsop[…] DdksADrelADsopT […]” 

Replace “sopT” with “spoT”. 

-Results. Lines 293-294. “Deletion of both DksA and (p)ppGpp […]”. 

This is not scientifically correct. A deletion applies to genes, so authors should use “dksA” instead of “DksA”.  Also, “deletion of (p)ppGpp” is incorrect.  Authors should either write “deletion of genes involved in (p)ppGpp production” or “mutant strains impaired for (p)ppGpp production”. 

-Results. Lines 328-329. “[…] cellular processes […] ribosome […], transporters (gntThisJ), RNA biosynthesis (trmDdusB), etc. ”

“Ribosome”and “transporters” are not cellular processes.  Authors should instead use “translation or ribosome biogenesis”, or “nutrients import”.  They should also refrain using “etc”, and use “and others” instead. 

-Results. Lines 329-330. “The primers were designed using Primer Premier 6 software […]. 

This information should be moved to the “Material and Methods” section. 

-Figure 6. The authors should identify the growth condition corresponding to each panel on the figure in addition of providing the information in the figure legend so the readers can quickly figure out the condition at glance.

-Results. Lines 403-405. “The decreased expression of these flagellar genes may be responsible for the decreased motility and flagellar formation ability of YENRS” 

The authors have to support this claim by referring to a figure from the manuscript (e.g. Figure 6B).

Round 2

Reviewer 2 Report

The authors properly addressed my comments, so I now recommend the publication of this manuscript.